# Diagnosis of Organizing Pneumonia with an Ultrathin Bronchoscope and Cone-Beam CT: A Case Report

**DOI:** 10.3390/diagnostics12112813

**Published:** 2022-11-16

**Authors:** Eleonora Casalini, Roberto Piro, Matteo Fontana, Laura Rossi, Federica Ghinassi, Sofia Taddei, Maria Cecilia Mengoli, Luca Magnani, Bianca Beghè, Nicola Facciolongo

**Affiliations:** 1Pulmonology Unit, Azienda Unità Sanitaria Locale—IRCCS di Reggio Emilia, 42123 Reggio Emilia, Italy; 2Respiratory Diseases Unit, Azienda Ospedaliero-Universitaria of Modena, 41121 Modena, Italy; 3Pathology Unit, Azienda Unità Sanitaria Locale—IRCCS di Reggio Emilia, 42123 Reggio Emilia, Italy; 4Rheumatology Unit, Azienda Unità Sanitaria Locale—IRCCS di Reggio Emilia, 42123 Reggio Emilia, Italy; 5Department of Surgery, Medicine, Dentistry and Morphological Sciences with Interest in Transplantation, Oncology and Regenerative Medicine, Faculty of Medicine and Surgery, University of Modena and Reggio Emilia, 41124 Modena, Italy

**Keywords:** organizing pneumonia (OP), primary Sjögren’s Syndrome (pSS), cone-beam computer tomography (CBCT), augmented fluoroscopy (AF), ultrathin bronchoscope (UTB), mini-forceps, small peripheral pulmonary lesions (PPLs)

## Abstract

Organizing pneumonia (OP) is a pulmonary disease histopathologically characterized by plugs of loose connective tissue in distal airways. The clinical and radiological presentations are not specific and they usually require a biopsy confirmation. This paper presents the case of a patient with a pulmonary opacity sampled with a combined technique of ultrathin bronchoscopy and cone-beam CT. A 64-year-old female, a former smoker, was admitted to the hospital of Reggio Emilia (Italy) for exertional dyspnea and a dry cough without a fever. The history of the patient included primary Sjögren Syndrome interstitial lung disease (pSS-ILD) characterized by a non-specific interstitial pneumonia (NSIP) radiological pattern; this condition was successfully treated up to 18 months before the new admission. The CT scan showed the appearance of a right lower lobe pulmonary opacity of an uncertain origin that required a histological exam for the diagnosis. The lung lesion was difficult to reach with traditional bronchoscopy and a percutaneous approach was excluded. Thus, cone-beam CT, augmented fluoroscopy and ultrathin bronchoscopy were chosen to collect a tissue sample. The histopathological exam was suggestive of OP, a condition occurring in 4–11% of primary Sjögren Syndrome cases. This case showed that, in the correct clinical and radiological context, even biopsies taken with small forceps can lead to a diagnosis of OP. Moreover, it underlined that the combination of multiple advanced technologies in the same procedure can help to reach difficult target lesions, providing proper samples for a histological diagnosis.

## 1. Introduction

Organizing pneumonia (OP) is a non-specific lung injury response that is histopathologically characterized by polypoid plugs of loose connective tissue in distal airways [1]. The radiological presentation of OP is polymorphous, but the predominant pattern shows areas of consolidations [2]. Less frequently, a computed tomography (CT) scan shows solitary or multiple nodules. These radiological finds can migrate or spontaneously regress [3].

The clinical presentation often mimics other disorders such as infections and cancer, frequently leading to a diagnostic delay [1]. The majority of patients usually present with a non-productive cough, a fever and/or exertional dyspnea [2].

Organizing pneumonia should be suspected based on the clinical and radiological presentations, but it generally requires a histopathological confirmation to exclude other possible etiologies including malignancies [4]. Various biopsy techniques have been used in order to diagnose this pathology, but endoscopic techniques such as transbronchial lung biopsies and transbronchial lung cryobiopsies have shown a high diagnostic yield [4,5]. The treatment usually consists of a prolonged corticosteroid therapy and disease relapses are common [2]. The presence of OP can be associated with several specific conditions, including infections, drug reactions, hypersensitivity pneumonitis, radiation therapy and connective tissue diseases (CTDs) [3]. When a specific etiology cannot be found, OP is classified as cryptogenic organizing pneumonia (COP); otherwise, it is labelled as secondary OP [6,7]. Among CTDs, OP has been mostly reported in association with rheumatoid arthritis, less frequently with secondary Sjögren’s and rarely with primary Sjögren’s Syndrome (pSS) [8,9].

Sjögren’s Syndrome (SS) is a chronic systemic autoimmune disease characterized by the hypofunction of the salivary and lacrimal glands [10]. This disease can be primitive (primary Sjögren’s Syndrome, or pSS) or associated with other systemic autoimmune diseases such as systemic lupus erythematous, rheumatoid arthritis or systemic sclerosis [11]. The clinical presentation typically includes mucosal dryness, especially involving the eyes and mouth, and up to 40% of patients can present with multiple organ involvement and systemic manifestations [11]. Pulmonary manifestations include airway abnormalities such as bronchiectasis or air trapping, reflecting constrictive obliterative bronchiolitis and interstitial lung disease (ILD) [12]. Other pulmonary manifestations such as pulmonary Non-Hodgkin lymphoma or amyloidosis are less frequent [13]. ILD represents the most frequent and serious pulmonary complication, with a prevalence of around 20% in patients with pSS [14]. pSS-associated ILD (pSS-ILD) can represent a late manifestation or it can develop years before the onset of pSS, with clinical symptoms including exertional dyspnea and a non-productive cough [15]. Lung function tests can be normal, but they generally show a restrictive ventilatory pattern characterized by a reduced total lung capacity (TLC) and diffusing capacity of the lung for carbon monoxide (DLCO) [16]. Less often, a mixed obstructive and restrictive pattern can be observed, secondary to both the airway and pulmonary parenchyma involvement [14,17]. High-resolution computed tomography (HRCT) represents the main imaging tool for evaluating pSS-ILD; it can show reticular abnormalities, honeycombing, ground-glass opacities, cysts, nodules, consolidations and bronchiectasis [18]. Non-specific interstitial pneumonia (NSIP) is the most frequent radiological and pathological pattern (41–45% of patients) [19], but other patterns such as usual interstitial pneumonia (UIP) or lymphocytic interstitial pneumonia (LIP) are common [18]. These patterns can be presented alone or combined [14]. An OP pattern is rare and with HRCT it typically appears as a subpleural consolidation or, less frequently, as a single mass or nodule. A multidisciplinary discussion between pulmonologists, radiologists, rheumatologists and pathologists is mandatory in the diagnostic work-up and definition of the therapeutic strategy [15,17]. A transbronchial lung biopsy (TBLB) can be useful as it allows the histological findings to be added to the diagnostic process. In the case of small peripheral lung lesions, new advanced techniques such as cone-beam computed tomography (CBCT) augmented fluoroscopy [20] or an ultrathin bronchoscope (UTB) [21] can be used.

The optimal therapeutic regimen for pSS-ILD has not yet been established. For patients with a symptomatic or progressive disease, corticosteroids are empirically used and are generally recommended for the treatment of OP (initial dose of 0.5–1 mg/kg of prednisone daily, gradually tapered). Other immunosuppressive drugs such as azathioprine, cyclophosphamide and mycophenolate mofetil can be considered in refractory cases or as a first-line regimen in association with steroids as steroid-sparing drugs [15,17].

We present here the case of a 64-year-old, ex-smoker female with a history of pSS-ILD, who presented with exertional dyspnea and a right lower lobe pulmonary opacity. The patient underwent a combined technique of cone-beam computerized tomography (CBCT) and ultrathin bronchoscopy to obtain samples for the histopathological diagnosis.

## 2. Case Presentation

A 64-year-old female former smoker (20 pack-years) was admitted to the hospital of Reggio Emilia (Italy) for exertional dyspnea and a dry cough without a fever. The history of the patient included obesity (body mass index: 31 kg/m^2^), arterial hypertension and a pSS-ILD characterized by an NSIP radiological pattern. She had been on treatment for pSS-ILD for five years, initially with high doses of corticosteroids and subsequently with azathioprine, because of a relapse of the symptoms after the tapering of the steroids. Azathioprine was suspended 18 months before the hospital admission due to the appearance of myalgia and an intolerance of the patient to it. The percutaneous oxygen saturation with room air was 98% at rest and a thorax auscultation revealed a few crackles at the basal part of the right lung. The lung function tests and diffusing capacity for carbon monoxide were normal. A SARS-CoV-2 real-time PCR test and the sputum cultures were negative. The laboratory blood tests were normal, except for D-dimer (4230 ng/mL; cut-off: < 500 ng/mL). No sign of a lower limb venous thrombosis was found at the physical examination and at the point of care ultrasound. The echocardiography was normal and a computed tomography pulmonary angiography (Somatom Definition Edge, Siemens Healthineers, Erlangen, Germany) excluded a pulmonary embolism, but a new right lower lobe pulmonary opacity (35 mm and 15 mm diameter) was detected (Figure 1).

Considering the characteristics of the consolidation, which was compatible with both inflammatory and neoplastic diseases, a histological analysis with a pulmonary biopsy was suggested during the multidisciplinary discussion. The lung lesion was difficult to reach with traditional bronchoscopy combined with a standard C-arm fluoroscopy guide due to the position, shape, consistency and size of the nodule. Considering the relevant risk of a pneumothorax from a percutaneous transthoracic needle biopsy, the patient did not accept it. Thus, cone-beam CT (CBCT)-guided bronchoscopy was proposed to the patient, who gave her informed consent.

The procedure was performed in a hybrid operating room with Discovery IGS 740 Cone-Beam CT equipment (GE Healthcare, Chicago, IL, USA) under general anesthesia and through an endotracheal tube. A standard flexible video bronchoscope with a 2.0 mm working channel (BF-H190, Olympus Medical Systems, Tokyo, Japan) was used to administer the local anesthesia and to explore the bronchial tree. The endoscopic procedure was then continued with an ultrathin bronchoscope with the aim of better reaching the target lesion and performing the lung biopsies. This bronchoscope (BF-XP190 Olympus Medical Systems, Tokyo, Japan) had an outer diameter of 3.1 mm (at the distal end) and a 1.7 mm working channel. A CBCT acquisition was initially performed to define the target position and the easiest path to reach it was through the bronchial tree. Images were obtained by a 5 s rotational acquisition around the patient, who was kept in apnea in order to avoid CBCT reconstruction breathing-related artifacts. Subsequently, an interventional pulmonologist and a radiologic technologist segmented the volumes of interest, consisting of the target lesion and the path to reach it, using dedicated CBCT-augmented planning and guidance software (ASSIST, GE Healthcare, Chicago, IL, USA). The target and endobronchial path CBCT-based 3D volumes were then automatically fused on two-dimensional X-ray fluoroscopy imaging to augment the live guidance (Figure 2; see also Appendix A). The images were adjusted from the workstation to maximize the live fluoroscopy and augmented guidance visualization during the procedure. The target was identified in the apical segment of the right lower lobe (RB6) and an ultrathin bronchoscope was used to better reach the correct bronchus leading to the lesion. A 1.5 mm diameter mini-forceps tool was then pushed to the lesion and a second CBCT acquisition was performed to confirm it reaching the location. Tissue sampling was performed using the mini-forceps under augmented fluoroscopy guidance with seven transbronchial lung biopsies (TBLB) from the target lesion site. Afterwards, broncho-alveolar lavage (BAL) was performed. No procedure-related complications were observed. The microbiological and cytological tests on the BAL fluid were negative for common bacteria, fungi and malignant cells. The histopathological specimens consisted of 7 small fragments (maximum diameter of 2.1 mm) of pulmonary parenchyma. Four biopsies (slide B, Figure 3) showed well-expanded alveolar spaces, but demonstrated only two non-diagnostic incidental features related to the smoking habit of the patient: (1) mild respiratory bronchiolitis with a degree of finely pigmented macrophages within the alveolar spaces and respiratory bronchioles; and (2) mild hyaline interstitial fibrosis. Three biopsies (slide A, Figure 4) showed marked artifactual atelectasis, giving the false impression of hypercellularity and rendering the pathological examination difficult. The paraffin block of slide A was re-cut to obtain 5 deeper levels. Although the microscopic evaluation was difficult, a few intra-alveolar polypoid plugs of loose connective tissue with several lymphocytes and histiocytes were detected. A final pathological diagnosis of lung parenchyma with a few foci of organizing pneumonia was made. In combination with the clinical and radiological findings, a final diagnosis of organizing pneumonia in pSS-ILD was formulated. The patient received oral prednisone 50 mg daily for two weeks; this was then gradually tapered. Her symptoms quickly improved and the lung opacity gradually decreased and resolved at the CT scan follow-up performed 18 weeks after the beginning of the therapy (Figure 5).

## 3. Discussion

Organizing pneumonia (OP) occurs in 4–11% of the cases of primary Sjögren’s Syndrome (pSS) [13,14,15]. The main finding from CT scans is the presence of patchy areas of airspace consolidation with a peripheral and/or peribronchovascular distribution, often bilateral and with air bronchograms. These opacities can migrate and spontaneously regress. Areas of ground-glass attenuation or fine centrilobular nodules can be observed in association with the consolidation areas. Less frequently, solitary or multiple nodules mimicking a malignancy can be detected [1,4,22]. The NSIP pattern can sometimes overlap with the OP pattern [4,7,15]. Histological analyses can reveal intraluminal polypoid plugs of loose connective tissue in the alveoli and bronchioles; the architecture of the lung is conserved and honeycombing is absent. Although the role of a pulmonary tissue biopsy is still debated, many experts consider it to be crucial to confirm an OP diagnosis [3]. A biopsy may be unnecessary if the clinical and radiological features are highly suggestive of OP, but a histological confirmation is certainly useful to rule out other potential diagnoses such as a malignancy. A multidisciplinary discussion involving pulmonologist, radiologist, rheumatologist and histopathologist experts in ILD may be helpful in the diagnostic framework and work-up [1,4,6].

A CT-guided transthoracic needle biopsy can be used to perform the biopsy of lesions, but it is burdened with the risk of a pneumothorax, which ranges from 5 to 35% [22,23,24,25]. A fluoroscopy-guided transbronchial lung biopsy (TBLB) is a safe diagnostic approach to obtain lung tissue with a reduced risk of a pneumothorax (0.02–4.9%) [26,27,28]. The diagnostic performance of a TBLB in peripheral pulmonary lesions is highly variable and dependent on the lesion size and on the presence of a “bronchus sign” [28,29,30,31]; nevertheless, it can be considered to be the initial diagnostic procedure in most scenarios. Even if samples obtained are small biopsy specimens and the diagnostic adequacy of a TBLB remains debated, in the presence of a typical clinical and radiological picture, a histological pattern compatible with OP is sufficient to make the diagnosis [4,5,32]. During bronchoscopy, BAL can be performed; in patients with OP, it usually shows a mixed cellularity pattern with an increment in lymphocytes, neutrophils and eosinophils. The combination of histological and cytological results obtained from a TBLB and BAL can be crucial in a differential diagnosis, especially in the case of infections or a malignancy.

A transbronchial lung cryobiopsy (TBLC) is a bronchoscopic technique that allows larger pieces of tissue to be obtained in comparison with a conventional forceps biopsy by using a cryoprobe. Recent European Respiratory Society guidelines [33] consider a TBLC to be an acceptable alternative to a surgical lung biopsy (SLB) for a histopathological diagnosis in patients with ILD of an undetermined type in experienced centers as they have a similar sensitivity and specificity [34,35]. Compared with a surgical lung biopsy, a TBLC is less invasive and less expensive; the main complications are a pneumothorax (9%) and bleeding (30%) and the technology is not widely available.

An ultrathin bronchoscope (UTB) with an outer diameter ≤ 3.5 mm can advance deeper into the bronchial tree than large-caliber bronchoscopes, allowing the fifth-generation bronchus level to be reached. Most conventional UTBs are equipped with a 1.2 mm working channel that only permits the use of < 1.2 mm diameter mini-forceps, providing specimens of a limited size. Nevertheless, new generation UTBs are equipped with a 1.7 mm working channel that allows the use of 1.5 mm forceps and a dedicated 1.4 mm radial probe endobronchial ultrasound (rEBUS) [36,37]. The size of specimens obtained using 1.5 mm forceps is smaller than those obtained with 1.8 mm or 1.9 mm standard forceps, but the diagnostic accuracy appears to be similar [21,38]. This tool is particularly useful in the diagnosis of small peripheral pulmonary lesions (PPLs), especially if used in combination with guiding methods that indicate the correct bronchial route to the lesion and its localization such as fluoroscopy [36,37], rEBUS [21,39,40] or cone-beam CT (CBCT) [41].

CBCT is a three-dimensional imaging technique that was developed in the early 2000s. It was recently introduced into the interventional pulmonology field and has already been used in many vascular and percutaneous procedures to improve targeting and treatment planning [42,43]. CBCT allows the target position to be defined and these 3D volumes can then be superimposed on two-dimensional X-ray fluoroscopy images in order to augment the live guidance (augmented fluoroscopy). This technique has been effectively used to improve the navigation success rate and diagnosis of peripheral lesions [43,44] with or without virtual bronchoscopic navigation (VBN) [44,45,46].

Combining the benefits of CBCT-augmented fluoroscopy with those of a UTB can potentially be very useful in the diagnosis of peripheral lung lesions. A prospective pilot study suggested an increase in navigation (25%) and diagnostic performance (20%) with an acceptable radiation dose by using CBCT-guided thin/ultrathin bronchoscopy associated with rEBUS and fluoroscopy [41]. Ali et al. [47] reported an overall diagnostic yield of 90% when performing a TBLB using a UTB, VBN and CBCT in 40 patients with peripheral pulmonary lesions (PPLs), whether they were solid or had mixed ground-glass opacity (GGO); these were 30 mm on the long axis with the presence of bronchus signs from the CT scans. This element in particular is considered to be important for the success of the procedure; patients without it could be recommended for other diagnostic strategies such as a surgical biopsy [47].

The bronchoscopic diagnosis of small peripheral lung lesions remains a diagnostic challenge; in the last years, many resources have been invested to develop new interventional pulmonology technologies to obtain effective tissue collections and to increase the diagnostic yield of conventional fluoroscopy-guided transbronchial biopsies. UTBs, rEBUS, CBCT-augmented fluoroscopy, VBN [40,48], electromagnetic navigation bronchoscopies (EBNs) [49] and robotic-assisted bronchoscopies [50,51,52] are the main new advances now available to improve the navigation to and achievement of the target lesion. The combination of different new bronchoscopic techniques in a multimodality approach can make the most out of their respective advantages and achieve a higher diagnostic yield in peripheral lung lesions [53].

In this clinical case, we used multiple advanced technologies in the same procedure in order to obtain the diagnosis. To the best of our knowledge, this is the first reported case of OP in pSS diagnosed through CBCT-augmented fluoroscopy with a TBLB performed through a UTB using 1.5 mm mini-forceps. Bronchoscopic tissue sampling was proposed by a multidisciplinary group, which judged the case to be suspicious for a malignancy in view of the radiological presentation and the smoking history of the patient. The use of a UTB allowed us to approach the target lesion more accurately compared with the use of a conventional bronchoscope. CBCT-augmented fluoroscopy was essential, not only for the correct localization of and navigation toward the nodule, but also to confirm being exactly on target during the sampling. The combined use of these two advanced techniques permitted us to obtain adequate histological specimens showing the OP pattern.

A modern interventional pulmonology unit should encourage the use of innovative technologies with dedicated staff in order to maximize the diagnostic performance, even when the case is more challenging and the lesion is difficult to reach. Adequate and constant training is necessary to ensure personnel are constantly updated and achieve satisfactory results.

## 4. Conclusions

In the diagnostic challenge of sampling peripheral pulmonary lesions, a great help has arrived from new technologies such as cone-beam CT-augmented fluoroscopy and ultrathin bronchoscopes. These tools ca be used for a diagnosis even in the case of non-neoplastic diseases and when mini-forceps are used.

## Figures and Tables

**Figure 1 diagnostics-12-02813-f001:**
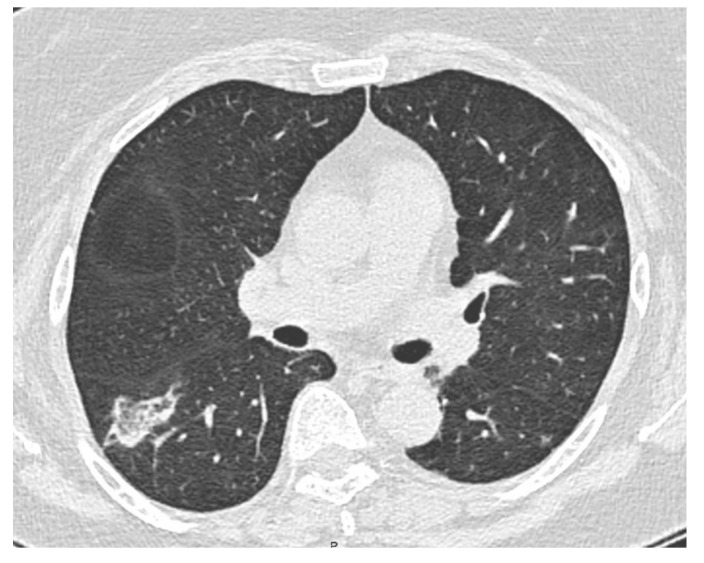
CT scan showing right lower lobe opacity.

**Figure 2 diagnostics-12-02813-f002:**
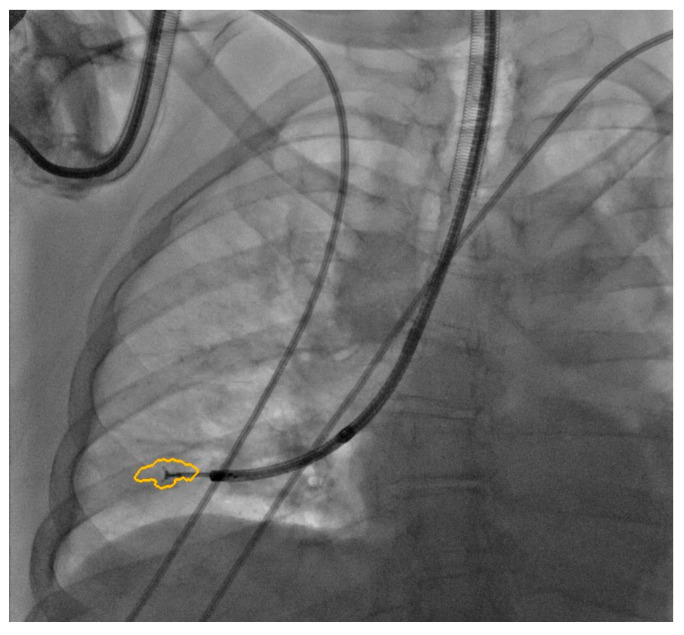
Augmented fluoroscopy.

**Figure 3 diagnostics-12-02813-f003:**
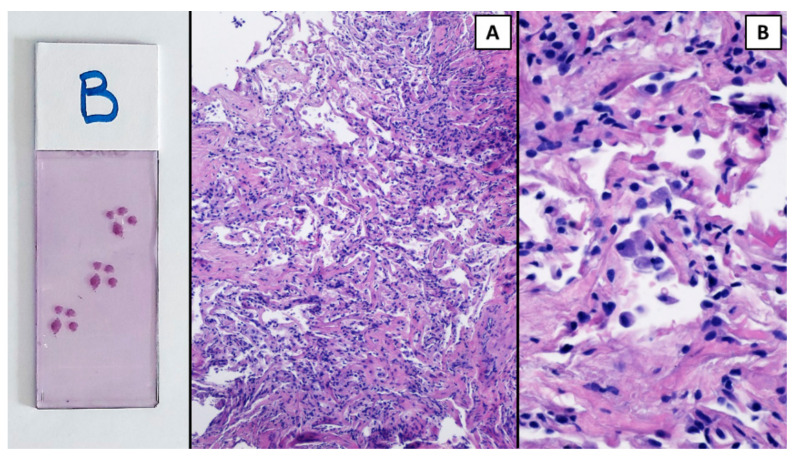
The pulmonary parenchyma in slide B was more preserved and expanded ((**A**), H&E 10×); however, it showed only mild hyaline interstitial fibrosis and respiratory bronchiolitis. H&E 40×. A collection of pigmented macrophages within the alveolar spaces at a higher power (**B**).

**Figure 4 diagnostics-12-02813-f004:**
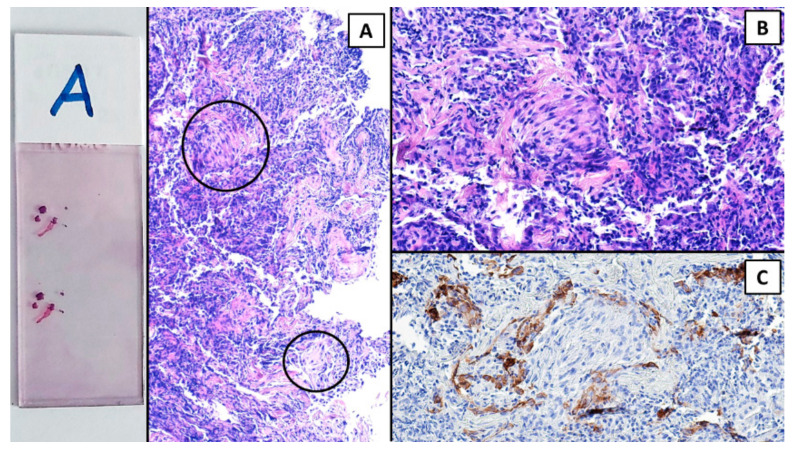
Hematoxylin and eosin (H&E). The 3 pulmonary fragments of slide A showed marked artifactual atelectasis; however, a few small but defined foci of organizing pneumonia emerged (within circles, (**A**)). (**B**) H&E, 20×. A myxoid plug of connective tissue intermingled with inflammatory cells. (**C**) The positive immunohistochemical reaction for cytokeratin in alveolar pneumocytes demonstrated that the focus of organizing pneumonia was within the alveolar space.

**Figure 5 diagnostics-12-02813-f005:**
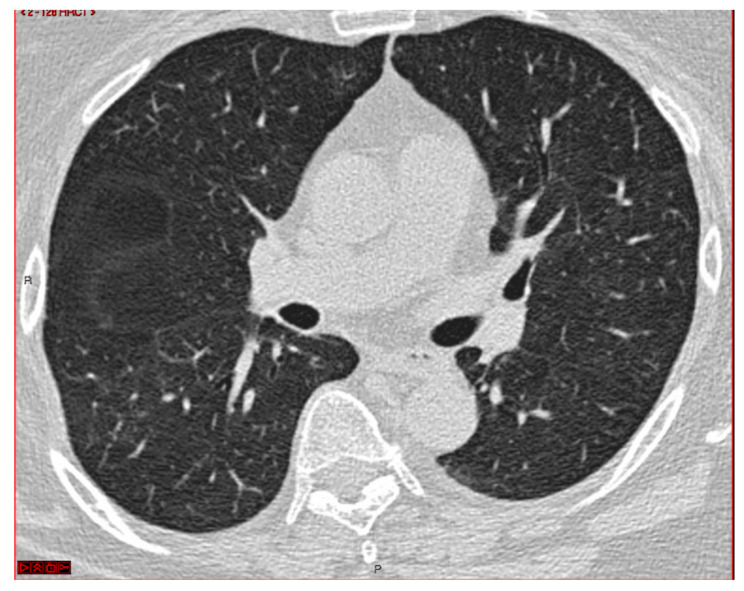
CT scan after therapy.

## Data Availability

Not applicable.

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
