# Peer review of "Diagnosis of Organizing Pneumonia with an Ultrathin Bronchoscope and Cone-Beam CT: A Case Report"

_diagnostics, 2022, doi:10.3390/diagnostics12112813_

Round 1

Reviewer 1 Report

This is a very interesting case report of OP in a pSS patient. The authors described the scheme of the diagnostic procedure in a very accessible and interesting way. As a reviewer, I have only 3 minor comments,

In the introduction section (lines 66-69) there should be a reference item documenting the frequency of pulmonary complications in pSS patients.

In the introduction section (lines 73-74), I suggest changing the text to .... restrictive ventilatory pattern characterized by reduced total lung capacity (TLC). TLC measurement is necessary to confidently diagnose restrictive changes; a reduction in FVC does not always equate to a diagnosis of restriction.

The case report itself lacks a thorough explanation of the reasons why transthoracic biopsy was not performed. From the CT slice presented, it appears that the lesion was not located deep in the lung parenchyma, there are no obvious signs of emphysema on the available CT section, SpO2 and PFT results were within normal limits, so why was the transthoracic biopsy abandoned?

Author Response

This is a very interesting case report of OP in a pSS patient. The authors described the scheme of the diagnostic procedure in a very accessible and interesting way. As a reviewer, I have only 3 minor comments,

Thanks to the Reviewer for her/his consideration. We are very pleased that she/he appreciates our work and found the manuscript very accessible and interesting.

In the introduction section (lines 66-69) there should be a reference item documenting the frequency of pulmonary complications in pSS patients.

Thank you: we added the reference, as suggested.

In the introduction section (lines 73-74), I suggest changing the text to .... restrictive ventilatory pattern characterized by reduced total lung capacity (TLC). TLC measurement is necessary to confidently diagnose restrictive changes; a reduction in FVC does not always equate to a diagnosis of restriction.

Thank you, we fully agree with the Reviewer and we corrected the sentence.

The case report itself lacks a thorough explanation of the reasons why transthoracic biopsy was not performed. From the CT slice presented, it appears that the lesion was not located deep in the lung parenchyma, there are no obvious signs of emphysema on the available CT section, SpO2 and PFT results were within normal limits, so why was the transthoracic biopsy abandoned?

Yes, the information reported by the Reviewer is correct. The Radiologist judged the transthoracic biopsy at a significant risk for pneumothorax and the patient, informed of that, refused it. Now we have better explained this aspect in the manuscript. Thans to the Reviewer for the comment.

Reviewer 2 Report

Dear Editor,

I would like to thank you the opportunity to review the manuscript entitled “Diagnosis of organizing pneumonia with ultrathin broncho-scope and cone-beam CT: a case report” and so contribute with this Journal.

The manuscript aimed to presents the case of a patient with pulmonary opacity sampled with a combined-technique of ultrathin bronchoscopy and cone-beam CT. The subject addressed is new, offers a scientific contribution and is of interest to the readers of the journal. Therefore, I would recommend publishing the manuscript.

See below my comments.

  1. Introduction
  • First and second paragraphs: I believe this is a writing style of the authors.  Authors must insert references that support the statements made in the two paragraphs.
  • Third and fourth paragraphs: The two paragraphs are long, with several sentences. Authors must reassign the references in order to support the information presented. 49 references were used.

2. Case presentation

- What device was used to acquire the tomographic images? What parameters were used to acquire the images?

3. Conclusions

- The conclusion must be redrafted. Authors should seek to conclude with the objective of the study as a reference.

Author Response

The manuscript aimed to presents the case of a patient with pulmonary opacity sampled with a combined-technique of ultrathin bronchoscopy and cone-beam CT. The subject addressed is new, offers a scientific contribution and is of interest to the readers of the journal. Therefore, I would recommend publishing the manuscript.

Thank you to the Reviewer for her/his very positive words. We are very pleased that she/he considered interesting the novelty of our contribution and that she/he recommended to publish the paper.

See below my comments.

  1. Introduction
  • First and second paragraphs: I believe this is a writing style of the authors.  Authors must insert references that support the statements made in the two paragraphs.

We agree with the Reviewer and we inserted the appropriate references, as suggested.

  • Third and fourth paragraphs: The two paragraphs are long, with several sentences. Authors must reassign the references in order to support the information presented. 49 references were used.

We thank the Reviewer for the comment and for the suggestion. We eliminated some unnecessary sentences and we reassigned the references to better support the information presented.

  1. Case presentation

- What device was used to acquire the tomographic images? What parameters were used to acquire the images?

CT scans were performed using a 128-slice scanner (Somatom Definition Edge, Siemens Healthineers, Erlangen, Germany), with the patient in the supine position, during end-inspiration. Scanning parameters were: tube voltage 120 KV, automatic tube current modulation, collimation width 1.25 mm, acquisition slice thickness 2.5 mm, and interval 1.25 mm.

The reference to the device is now added in the text.

  1. Conclusions

- The conclusion must be redrafted. Authors should seek to conclude with the objective of the study as a reference.

Thanks to the Reviewer for help us improve the paper with this suggestion. We have completely redrafted the conclusion, in order to focus it to the objective of the study.

Reviewer 3 Report

- the name of the hospital the patient was admitted in is not necessary 

- what is the benefit of using such equipments in the diagnosis if the course of the treatment doe not 

Author Response

- the name of the hospital the patient was admitted in is not necessary 

Thank you for the suggestion: we deleted the name of the hospital.

- what is the benefit of using such equipments in the diagnosis if the course of the treatment doe not 

Thank you for the answer. We think that reaching the diagnosis was very important because:

  • the differential diagnosis of the lung consolidation included other disease (e.g. malignancies)
  • once the diagnosis of organizing pneumonia was obtained, the patient began a steroid treatment